# Assessing the impact of lateral flow testing strategies on within-school SARS-CoV-2 transmission and absences: A modelling study

**Trystan Leng**[1,2¤] *, **Edward M. Hill**[1,2], **Robin N. Thompson**[1,2], **Michael J. Tildesley**[1,2], **Matt J. Keeling**[1,2], **Louise Dyson**[1,2]

**1** The Zeeman Institute for Systems Biology & Infectious Disease Epidemiology Research, School of Life Sciences and Mathematics Institute, University of Warwick, Coventry, United Kingdom, **2** JUNIPER – Joint UNIversities Pandemic and Epidemiological Research, https://maths.org/juniper/

¤ Current address: Department of Infectious Disease Epidemiology, Imperial College London, London, United Kingdom
* trystan.leng@imperial.ac.uk

**Data Availability Statement:** There are no primary data in the paper; the source code and data used to produce the results and analyses presented in this

## Abstract

Rapid testing strategies that replace the isolation of close contacts through the use of lateral flow device tests (LFTs) have been suggested as a way of controlling SARS-CoV-2 transmission within schools that maintain low levels of pupil absences. We developed an individual-based model of a secondary school formed of exclusive year group bubbles (five year groups, with 200 pupils per year) to assess the likely impact of strategies using LFTs in secondary schools over the course of a seven-week half-term on transmission, absences, and testing volume, compared to a policy of isolating year group bubbles upon a pupil returning a positive polymerase chain reaction (PCR) test. We also considered the sensitivity of results to levels of participation in rapid testing and underlying model assumptions. While repeated testing of year group bubbles following case detection is less effective at reducing infections than a policy of isolating year group bubbles, strategies involving twice weekly mass testing can reduce infections to lower levels than would occur under year group isolation. By combining regular testing with serial contact testing or isolation, infection levels can be reduced further still. At high levels of pupil participation in lateral flow testing, strategies replacing the isolation of year group bubbles with testing substantially reduce absences, but require a high volume of testing. Our results highlight the conflict between the goals of minimising within-school transmission, minimising absences and minimising testing burden. While rapid testing strategies can reduce school transmission and absences, they may lead to a large number of daily tests.

## Author summary

During the COVID-19 pandemic, a range of measures have been implemented to reduce transmission in schools. If an infected pupil is detected, one approach involves 'isolating'

manuscript are available from https://github.com/tsleng93/SchoolReopeningStrategies and on Zenodo: https://doi.org/10.5281/zenodo.6540077.

**Funding:** MJK, RNT, MJT and LD were been supported by the Engineering and Physical Sciences Research Council through the MathSys CDT [grant number EP/S022244/1]. EMH, MJK, LD and MJT were supported by the Medical Research Council through the COVID-19 Rapid Response Rolling Call [grant number MR/V009761/1]. TL, MJK, LD and MJT were supported by Medical Research Council through the JUNIPER modelling consortium [grant number MR/V038613/1]. MJK was supported by the National Institute for Health Research (NIHR) [Policy Research Programme, Mathematical \& Economic Modelling for Vaccination and Immunisation Evaluation, and Emergency Response; NIHR200411]. The funders had no role in study design, data collection and analysis, decision to publish, or preparation of the manuscript.

**Competing interests:** The authors have declared that no competing interests exist.

their contacts, who must then remain at home and not attend school. However, this may lead to high levels of pupil absences, which in turn may impact educational attainment. An alternative approach involves testing the contacts of infected individuals each day, who are allowed to attend school if they test negative. This can be achieved using lateral flow device tests, which can be taken at home and return a result in under thirty minutes. In this paper, using an individual-based model, we compare the impact of strategies that use lateral flow device tests to rapidly test secondary school pupils to strategies that require the contacts of infected individuals to isolate. We find that a strategy that combines testing pupils regularly with testing the contacts of identified infected individuals for a period of seven days has the potential to keep both transmission and absences low. However, such a strategy requires a high number of tests. Our results demonstrate the conflict between the competing aims of minimising transmission, minimising absences, and minimising test burden.

## Introduction

To control the spread of SARS-CoV-2, the causative agent of COVID-19 disease, unprecedented restrictions have been placed upon people's daily lives. In the UK, these non-pharmaceutical interventions (NPIs) have included the wearing of face coverings, social distancing, the prohibitions of households mixing socially, the restriction of a range of leisure activities, and the closure of educational establishments, workplaces, pubs, and restaurants. Prior to the development and approval of vaccines, which have become central to controlling SARS-CoV-2, NPIs were required to reverse the growth in infection during the first and subsequent waves. Even since the roll-out of vaccines, NPIs have continued to play an important role in controlling transmission.

While these NPIs together can be effective at controlling the spread of infectious diseases, their implementation has come with significant societal and economic costs [1–3]. In particular, the closure of schools puts educational outcomes at risk, especially for disadvantaged pupils, with existing inequalities and attainment gaps being exacerbated [4, 5]. School closures have a particularly adverse impact on vulnerable children due to reduced access to essential services [6], and impair the physical and mental health of many children [7, 8]. The closure of schools has also placed an additional strain on parents, and the burden of care during lockdowns has exacerbated gender inequities [9]. Owing to these harms, the UK government expressed that keeping schools open was a priority in their late 2020 plans [10].

Evidence from a range of sources suggests that children are, in general, only mildly affected by the disease and have low mortality rates [11, 12]. This is reflected by the fact that by 1st December 2021 there had been 98, 599 people who had died in hospitals in England and had tested positive for COVID-19, but only 76 of those were in the 0–19 year age group [13]. However, the role of children in the spread of SARS-CoV-2, particularly in the school setting, is complex. A meta-analysis of 31 empirical studies concluded that there is weak evidence that school-aged children play a smaller role in SARS-CoV-2 transmission than adults [14], and analyses of community testing data [15] and school absences data [16] have suggested that cases among school-aged children followed similar trajectories to those in the adult population in England in 2020 (a period when schools implemented strict control measures). Another meta-analysis of 37 contact-tracing and population studies concluded there were no differences in SARS-CoV-2 transmission between children and adults, but found markedly lower secondary attack rates from children in the school setting compared to the household setting [17]. Other studies have suggested that a suite of within-school control measures implemented

in tandem can mitigate within-school transmission [18, 19]. On the other hand, when limited school control measures have been implemented, substantial school outbreaks have been observed in other contexts [20, 21], and in England a surge in cases amongst children was observed after schools reopened in August 2021 [22]. Within-school transmission will also be influenced by other factors, including community prevalence [23] and the transmissibility of circulating strains.

Given the societal benefits of keeping schools open, a variety of strategies aiming to minimise transmission within the school setting have been considered and applied. In the UK, a range of measures have been introduced to schools over the course of the pandemic [24], including socially distanced spacing between desks, restricting the movement of pupils within the school, restricting social mixing between year groups, mandatory mask wearing, as well as an increased emphasis on hand washing and general hygiene measures. From September 2020 to July 2021, to halt chains of transmission within a school, upon a pupil receiving a positive polymerase chain reaction (PCR) test, other pupils who have been in close contact with the positive case had to isolate for 10 days (14 days originally). Originally, these 'close contacts' typically referred to an entire year group bubble, although some schools have since implemented more targeted approaches [25]. While the isolation of close contacts is expected to have been effective at reducing transmission, it has also led to a considerable number of school days missed throughout the term. These absences have an impact on children's education and the ability to assess pupils fairly, and undermine the benefits of keeping schools open.

Since March 2021, rapid coronavirus testing using lateral flow device tests (LFTs) has been a part of England's secondary school control strategy, with secondary school pupils and staff strongly encouraged to take two LFTs a week [24]. LFTs can be taken at home and return a result in under 30 minutes, making them ideal candidates for regular mass testing. From March 2021 to July 2021, this policy of mass testing was coupled with an isolation of close contacts policy. However, it was originally envisaged that LFTs could be used as an alternative to the isolation of close contacts [26]. In January 2021, it was proposed that testing would be provided (as a pair of LFTs) for all secondary pupils and staff prior to a return to face-to-face teaching. After this, staff would be tested once a week on an ongoing basis. Additionally, should a pupil receive a positive test, all close contacts of the pupil would be tested using an LFT for the next seven days, only attending school on receipt of a negative test. Positive tests identified during this period would trigger a further round of testing, until no new cases were been identified for a period of seven days. By identifying and isolating asymptomatic and presymptomatic individuals, it was hoped that such a strategy, referred to as *serial contact testing*, would be effective in controlling transmission within schools while leading to considerably fewer absences than would occur through isolating entire bubbles. However, the roll-out of this strategy was paused after the emergence of the more transmissible Alpha (B.1.1.7) variant [27]. To combat the spread of the highly transmissible Omicron variant, as of December 2021, serial contact testing has been implemented more widely in the UK for vaccinated contacts or contacts under the age of 18, who are not legally required to self-isolate [28].

While LFTs are less sensitive than PCR tests [29], a range of empirical [30, 31] and modelling [32–34] studies have demonstrated LFTs can play a valuable role in reducing SARS-CoV-2 transmission in a range of contexts. In particular, a cluster randomised trial of secondary schools in England explored the efficacy of serial contact testing, which suggested that such a strategy is non-inferior to the isolation of close contacts in reducing within-school transmission [31]. While the low sensitivity of LFTs observed in a field study of asymptomatic people in the city of Liverpool [35] led some early commentators to question their use in mass testing [36], more recently it has been argued that the results from the field study had been misinterpreted; the suggestion is LFTs are in fact more suitable for mass testing strategies as PCR tests

often return positive results for post-infectious individuals [37]. Indeed, LFTs have formed part of the UK's response in a variety of different contexts [24, 38, 39], and have been available for free to the population at large since 9th April 2021.

Mathematical modelling studies are vital tools to determine the efficacy of different control measures. Individual-based models of SARS-CoV-2 transmission in schools have elucidated the potential impact of a range of control measures, such as the cohorting of pupils [40, 41], mask-wearing [42], and PCR-based control strategies [43]. Other studies have used individual-based models to consider the impact of strategies involving LFTs on transmission within primary schools [44, 45]. In this study, we define an individual-based model of a secondary school to assess the potential impact of a range of potential strategies involving the use of LFTs on within-school transmission and absences, and in particular we consider the impact of serial contact testing.

To assess the potential impact of a range of isolation and testing strategies on transmission and absences, we created an individual-based model of a secondary school formed of exclusive 'year group bubbles'. We considered a range of strategies implemented or considered at various stages of the COVID-19 pandemic in England. By performing computational simulations corresponding to infection spread over the course of a school half-term of seven weeks, we compared the impact of strategies involving rapid testing using LFTs to a strategy of isolating year group bubbles on the total number of infections within schools, the number of school days missed per pupil and the number of LFTs taken per pupil. By doing so, we offer an assessment of the relative merits of a range of strategies involving LFTs compared to an isolation of year groups strategy. Further, we considered the impact of levels of pupil participation in lateral flow testing, and considered the sensitivity of our results to underlying model assumptions. By doing so, we identify factors likely to have the largest impact on the success of strategies involving the use of LFTs.

## Methods

In this study, we used a discrete-time stochastic individual-based model, with a daily time step, to simulate the spread of infection within a secondary school over a half-term of seven weeks. In our simulations, schools consisted of five year groups, with each year group containing 200 pupils, equivalent to a secondary school without a sixth form (ages 11–16, the inclusion of additional year groups representing a sixth form does not qualitatively change results). We provide a summary of our baseline and sensitivity assumptions in Table 1.

### Model initialisation

We initialised population prevalence and immunity ($I_{init}$ and $R_{init}$, respectively) to reflect the epidemiological situation in England in January 2021, when the introduction of serial contact testing was being considered. Specifically, we assumed that $I_{init}$ = 2% in line with contemporaneous community surveillance surveys [46]. We set $R_{init}$ = 20% based on an estimate from December 2020 that approximately 12% of the population in England would have tested positive for antibodies to SARS-CoV-2 from a blood sample [47], with an expectation the true proportion previously infected would be higher owing to waning of detectable antibodies [48]. Within each school, the initial number of susceptible, infected, and immune pupils within each school were sampled from the multinomial distribution Multinomial(1000, $[1 - I_{init} - R_{init}, I_{init}, R_{init}]$).

### Transmission

We assumed that simulated schools implemented a 'bubbling' policy at the level of year groups. In our baseline scenario, we assumed exclusive and effective year group bubbles, meaning that

**Table 1. Description of model parameters, our baseline parameterisation and the alternative values considered in the univariate sensitivity analysis.**

| Description | Baseline | Sensitivity | Source |
|---|---|---|---|
| LFT sensitivity | 30 day positive test probability profile for symptomatics<br>Faster decay in positive test probabilities for asymptomatics (Fig B in S1 Text) | 95% credible intervals from Hellewell *et al.* [49] for symptomatics<br>Transformed 95% credible intervals from Hellewell *et al.* [49] for asymptomatics | [49, 50] |
| PCR sensitivity | As above | As above | [49, 50] |
| within-school transmission, $K$ | Unif(1,5) | Baseline—$K = 3$<br>Low—$K = 1$<br>High—$K = 5$ | Assumption |
| External daily probability of infection to each pupil, $\epsilon$ | 0.001<br>(such that 10% of pupils are infected over a half term) | Low—$0.5 \times \epsilon$<br>High—$2 \times \epsilon$ | Assumption |
| Relative probability of transmission since day of infection, $\Gamma_I(d)$ | $\Gamma(5.62, 0.98)$ | - | [51] |
| Incubation period (time until symptom onset) | $\Gamma(5.807, 0.948)$ | - | [52] |
| % of pupils symptomatic | Unif(12,31) | Baseline—21.5%<br>Low—12%<br>High—31% | [53] |
| Relative infectiousness of asymptomatic individuals, $a$ (%) | Unif(30, 70) | Baseline—50%<br>Low—30%<br>High—70% | [54, 55] |
| Initial population level immunity, $R_{init}$ | 20% | Low—10%<br>High—30% | [47] |
| Initial population level prevalence, $I_{init}$ | 2% | - | [46] |
| Interaction between year groups, $\alpha$ | 0—i.e. 100% of within-school infections are within-year | 1—i.e. 20% of within-school infections are within-year | Assumption |

there was no transmission between year groups, but pupils mix randomly within year group bubbles. On school days, infected pupils transmitted infection to other pupils within their year with a probability dependent on (i) the number of days since they were infected, (ii) whether the pupil was asymptomatic, and (iii) a baseline within-school transmission parameter, $K$. An infected pupil's probability of transmission since day of infection progressed according to an infectivity profile for SARS-CoV-2 [56], derived from data from known source-recipient pairs [51], with an assumed incubation period distribution under the assumption that the generation time and incubation period are independent. Specifically, we assumed the infectivity profile of an infected pupil's relative probability of transmission since the day of infection ($\Gamma_I(d)$) was given by a gamma distribution with shape 5.62 and scale 0.98. Informed by previous studies into the levels of asymptomatic infection within age-groups, we assumed that 12–31% develop symptoms over the course of their infection [53, 57], with the rest of the school population remaining asymptomatic. Estimates of the relative infectiousness of asymptomatic individuals vary, though there are indications they may be less infectious than symptomatic individuals [54, 55]. Accordingly, we assumed that asymptomatic pupils were 30–70% as infectious as those that develop symptoms, denoting the relative infectiousness of asymptomatic pupils as $a$. Symptomatic pupils developed symptoms on a day drawn from a discretisation of a Gamma distribution with shape 5.807 and scale 0.948 [52] (see S1 Text), corresponding to a mean time to symptom onset of 5.5 days. We assumed that the relative probability of transmission of an individual and the time to symptom onset were independent, though in reality these factors likely influence one another [56, 58, 59]. Under our assumptions, approximately 50% of infectiousness occurred during an individual's presymptomatic infection phase. The level of onward transmission within-school remains unclear, and is likely to be influenced by a variety

of factors, including the success of other within-school social distancing measures and the epidemiological characteristics of the dominant strain of SARS-CoV-2 in circulation in the local area. Because of this, we considered a wide range of levels of within-school transmission $K$. Over the range considered, between 24% and 76% of infections occurred through pupil-to-pupil transmission under an isolation of year group bubbles strategy. Letting $N$ denote the number of pupils in a year group, the probability of transmission on day $d$ between an infectious pupil $i$ infected on day $d_0$ to a susceptible pupil $j$ in the same year group is given by:

$$\tau(i,j) = \begin{cases} \dfrac{K \times \Gamma_I(d - d_0)}{N - 1} & \text{, if } i \text{ is symptomatic and both pupils attend school} \\[2mm] \dfrac{a \times K \times \Gamma_I(d - d_0)}{N - 1} & \text{, if } i \text{ is asymptomatic and both pupils attend school} \\[2mm] 0 & \text{, if } i \text{ or } j \text{ does not attend school} \end{cases} \quad (1)$$

We did not explicitly model teachers, siblings, or external contacts. The impact of teachers and siblings can be indirectly captured by assuming some degree of transmission between year groups (S1 Text), a scenario considered in our sensitivity analysis (S2 Text). We assumed that the impact of external contacts on transmission could be captured by a constant daily probability external of infection to pupils, $\epsilon$, chosen to satisfy an average of 10% of pupils becoming infected by the end of the half-term under an isolation of year groups policy. When isolating, we assumed that individuals adhered and effectively isolated. After 15 days, we assumed individuals were no longer infectious and recover with immunity. For a more detailed summary of the simulation model, see S1 Text.

## Testing

Upon symptom onset, infected pupils underwent a PCR test. Pupils self-isolated until they received a test result, and we assumed that pupils received a result two days after taking a test. Those receiving a negative result returned to school the day after receiving their result, while those testing positive entered isolation for ten days. Pupils who tested positive using a LFT entered isolation, with the outcome of a confirmatory PCR test then determining whether the pupil remained in isolation or was released from isolation. Accordingly, we assumed that identified infected pupils did not transmit infection on the day they were tested. The impact of transmission from infected pupils on test days, corresponding to a situation where pupils are tested in school and effective isolation of identified positive cases is not possible, is considered in S3 Text. We used previously estimated LFT and PCR test probability profiles for symptomatic individuals [49]. For asymptomatic individuals, we assumed that the probability of testing positive was equal to that of symptomatic individuals until peak positive test probability, but then decayed more rapidly (S1 Text). We assumed a PCR test specificity of 100%, in line with studies observing that PCR tests rarely return false positive results [60, 61]. We assumed LFT specificity to be 99.97%, in line with estimates obtained from analyses of LFTs taken in secondary schools in England [62, 63]. In the following section we describe the six school reopening testing strategies considered.

## Reopening strategies

By simulating different school return strategies with the same set of parameter values and the same set of pregenerated random numbers, we could directly compare the impact each strategy had for each simulation. Specifically, random numbers determining the outcome of each transmission or testing event each day were pregenerated for each parameter set. We produced 10,000 simulation replicates for six strategies. Under all strategies involving the use of

LFTs ((ii)-(v)), participating pupils were also tested twice in the week before returning to school. Any positive cases that were identified via an LFT took a confirmatory PCR test, with the need to continue in isolation dictated by the result of that test.

(i).  **Isolation of year group bubbles**. Under this strategy, upon the identification of a case (through a symptomatic pupil seeking a PCR test), all pupils within a year group bubble were placed in isolation for ten days from the day after identification of the positive case. This approach was widely adopted by secondary schools in England from September 2020 to December 2020. If pupils tested positive while their bubble was placed in isolation, the period of isolation for the year group bubble was not extended.

(ii).  **Twice weekly mass testing and isolation of year group bubbles**. In this strategy, after the school reopened, modelled pupils attending school were tested twice a week three days apart. Three year groups were tested on Mondays and Thursdays, while two year groups were tested on Tuesdays and Fridays. Upon identification of a case, all pupils within a year group bubble were placed in isolation for ten days from the day after identification of the positive case. A combination of isolation of close contacts alongside twice weekly mass testing was implemented in secondary schools in England from March 2021 to July 2021.

(iii).  **Serial contact testing**. On the identification of a case (following confirmation by PCR), pupils attending school within a year group bubble were tested daily using LFTs for seven days from the day after identification of the positive case. The period of serial contact testing was reset if another pupil in the year group bubble returned a positive LFT result. Pupils testing positive to an LFT underwent a confirmatory PCR test—if this was negative, the period of serial contact testing was only extended until the day the negative result was returned. School testing did not occur at weekends—for serial contact testing days that fell on a weekend, pupils were assumed to be in isolation. This policy was considered as a school reopening strategy for secondary schools in England in January 2021.

(iv).  **Twice weekly mass testing and serial contact testing**. After the school reopened, modelled pupils attending school were tested twice a week three days apart. Three year groups were tested on Mondays and Thursdays, while two year groups were tested on Tuesdays and Fridays. Identification of positive cases, either through twice weekly mass tests, or symptomatic pupils seeking a PCR test, triggered the onset of serial contact testing as in (iii).

(v).  **Twice weekly mass testing**. After the school reopened, modelled pupils attending school were tested twice a week three days apart. Three year groups were tested on Mondays and Thursdays, while two year groups were tested on Tuesdays and Fridays. Any positive cases were isolated for a period dependent on the result of their confirmatory PCR test. However, positive tests did not result in any further school testing, nor in the isolation of year group bubbles. This approach was widely adopted by secondary schools in England from September 2021 to December 2021.

(vi).  **No school-level testing or isolation of year group bubbles**. Finally, we considered a situation in which a school implemented a control strategy involving no testing and no isolation of year group bubbles. Other than symptomatic individuals isolating when seeking a PCR test, and isolating for 10 days from the test upon receiving a positive test, no testing measures to control the spread of within-school infection were included. We considered this situation to enable assessment of the effectiveness of other strategies at reducing within-school transmission.

### Participation in lateral flow testing

Our main analysis assumed that all pupils participated in the school's control strategy. We did so to explore the impact of such strategies when optimally implemented. However, in practice it is unlikely that all pupils would comply with a school's control strategy. Accordingly, we also considered the impact of pupil participation in lateral flow testing. To account for this, we assumed that some proportion of pupils did not agree to participate in lateral flow testing throughout the half term. For strategies (iii) and (iv), i.e. those involving serial contact testing, pupils who opted out of lateral flow testing were required to isolate for ten days following the last contact with the positive case. For strategy (v), pupils who opted out were allowed to remain in school. We assumed that pupils who opted out of serial contact testing also opted out of twice weekly mass testing, and vice versa.

### Sensitivity

We assessed the sensitivity of results obtained to the underlying modelling assumptions (S2 Text). We explored the impact of altering the initial level of immunity among pupils (S4 Text), altering the assumed level of within-school transmission (S5 Text), the impact of cohorting year groups into smaller 'bubbles' (S6 Text), and the impact of LFT user error (S7 Text). For each scenario we generated 2000 simulation replicates. We performed the reopening strategy model simulations, sensitivity analysis and visualisation of results using MATLAB 2021a.

## Results

### Infections

Control strategies (i)-(v) reduced infections within school in the following order (from least to most effective): (iii)—serial contact testing, (i)—isolating year group bubbles, (v)—twice weekly mass testing (without further control measures), (iv)—twice weekly mass testing combined with serial contact testing, (ii)—twice weekly mass testing combined with isolation. In particular, for 82.0% of the 10,000 simulation replicates an isolation strategy without regular mass testing caused a greater reduction in infection than a serial contact testing strategy. Compared to isolating year group bubbles, a strategy of twice weekly mass testing (without further control measures) was more effective at reducing transmission (61.2% of simulations vs isolation of year group bubble strategy). A strategy that combined twice weekly mass testing with serial contact testing was yet more effective, resulting in fewer infections than an isolation strategy in 96.3% of simulations. However, a strategy of twice weekly mass testing combined with isolation was clearly the most effective strategy, leading to fewer pupils infected in 99.9% of simulations compared to a strategy combining twice weekly mass testing with serial contact testing (Fig 1A).

As part of our model setup, for the isolation of year group bubbles strategy (strategy i), we set the average number of pupils infected during the half-term at 10%. Owing to the wide range of levels of within-school transmission considered, alongside stochasticity, the incidence (over the duration of the half-term) varied considerably between simulations (95% prediction interval (PI): 5.9–17.8%). Considerably lower levels resulted from a strategy combining twice weekly mass testing with isolation (5.6%, 95% PI: 4.1–7.4%), or from a strategy combining twice weekly mass testing with serial contact testing (7.2%, 95% PI: 5.2–10.0%). We observed comparable half-term incidence distributions (compared to an isolation of year group bubble control strategy) for strategies of twice weekly mass testing (9.3%, 95% PI: 5.9–16.0%) and serial contact testing (10.7%, 95% PI: 6.3–19.1%). Without control measures, a mean of 16.6% of all pupils had been infected by the end of the half-term (95% PI: 6.8%-42.9%), and large

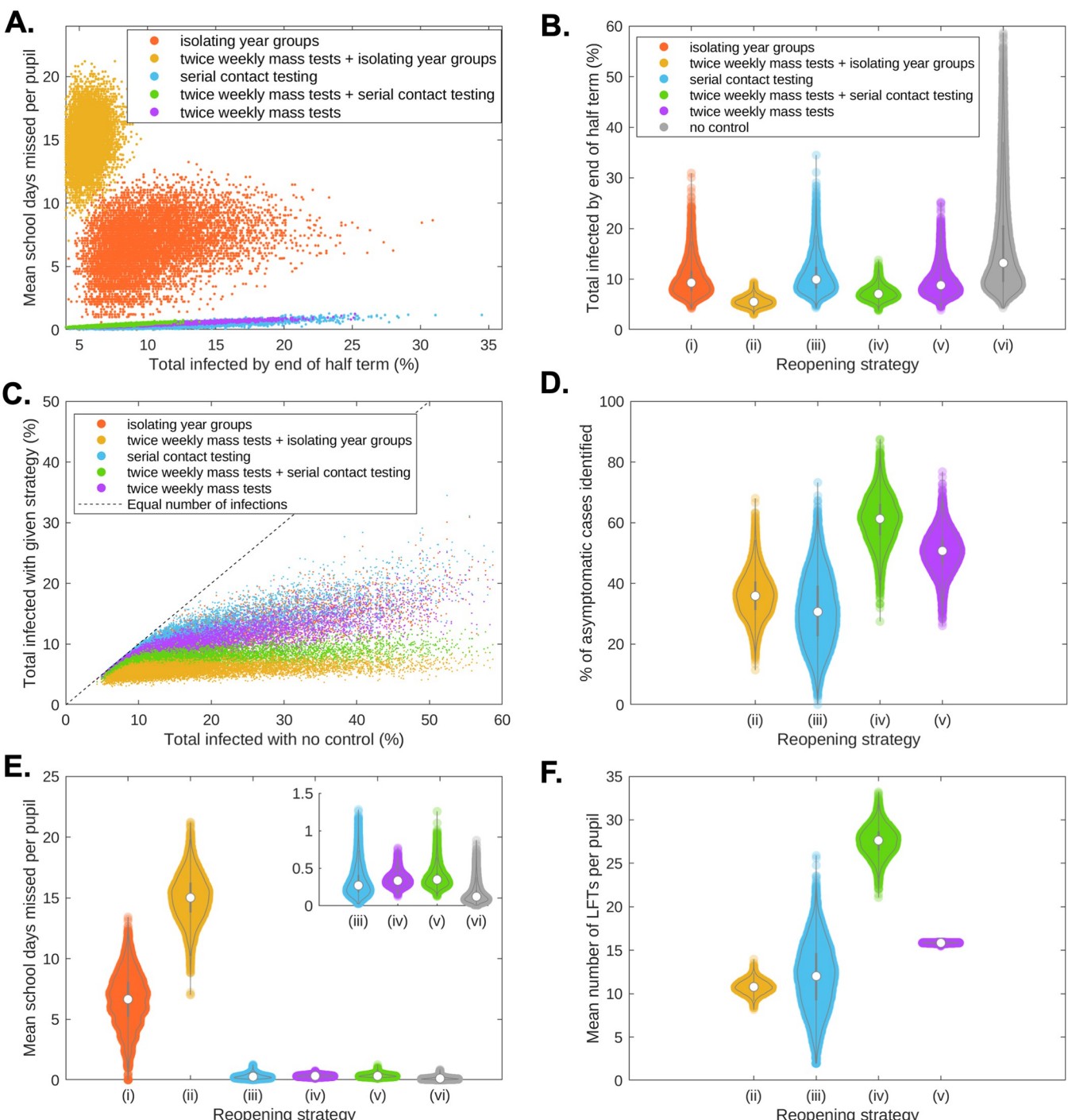

**Fig 1. The trade-off between transmission, absences, and testing volume.** (a) Relationship between total infections and school days missed for an isolation of year group bubbles strategy (orange), twice weekly mass testing combined with isolation of year group bubbles (yellow), serial contact testing (blue), twice weekly mass testing combined with serial contact testing (green), and twice weekly mass testing alone (purple). (b) Percentage of school pupils infected during the course of the half-term.(c) Comparison of the total number of infections by the end of the half-term by the five considered control strategies (y-axis) against the total number of infections by the end of the half-term with no infection control measures (x-axis). The dashed line corresponds to parity in the total amount of infection using no infection controls and the given control strategy. (d) For rapid testing strategies, the percentage of asymptomatic cases that had been identified through rapid testing by the end of the half term. (e) Violin plots of the mean number of school days missed per pupil within simulated schools. The inset plot shows strategies (iii)-(vi). (f) For rapid testing strategies, violin plots of the mean number of LFTs taken per pupil within simulated schools. Results produced from 10,000 simulations. In all violin plots, the circle marker denotes the median and the black bars the 50% prediction intervals.

outbreaks occurred much more frequently than under any strategies implementing control measures (Fig 1B). In particular, while some simulation replicates led to very large outbreaks strategies when no control strategy was implemented, combining twice weekly mass testing with either isolation or serial contact testing kept the numbers of cases low over the course of the half-term (Fig 1C).

We found that mass testing prior to the start of term initially reduced mean prevalence within schools, through transmission chains being interrupted by identifying asymptomatic and presymptomatic individuals (Fig 2A). Under a strategy of serial contact testing, prevalence increased to a higher level than would be obtained under an isolation of year group bubbles strategy by the second week of the half-term, with the higher prevalence then sustained throughout the rest of the half-term (Fig 2A, blue curve). A strategy of twice weekly mass testing alone maintained a lower level of prevalence than an isolation of year groups strategy until week 4, after which they had similar prevalence levels (Fig 2A, purple and orange curves). Combining twice weekly mass testing with either serial contact testing or isolating year group

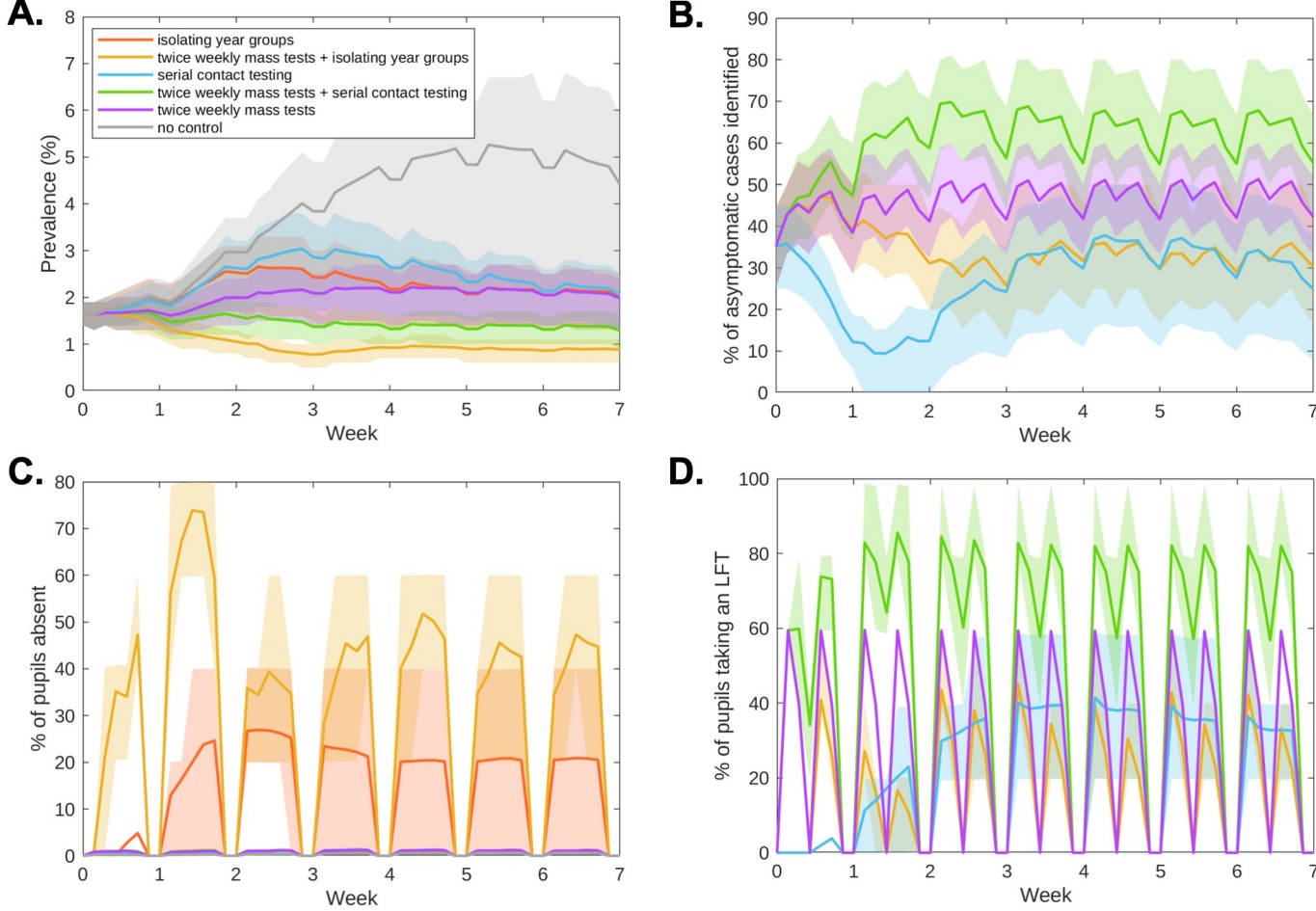

**Fig 2. Infection, absences, and testing over the duration of the school half-term.** We display timeseries of **(a)** prevalence, **(b)** the percentage of currently infectious asymptomatic individuals identified for reopening strategies involving within-school testing. We display timeseries of **(c)** the percentage of pupils absent and **(d)** percentage of pupils tested throughout the half-term. Solid line traces correspond to the mean value attained on each daily timestep and shaded envelopes represent the 50% prediction intervals (these regions contain 50% of all simulations at each timepoint). The strategies displayed are: no control (grey), twice weekly mass testing (purple), serial contact testing (blue), isolating year group bubbles strategy (orange), combined twice weekly mass testing and serial contact testing (green), combined twice weekly mass testing and isolation of year group bubbles (yellow). Results produced from 10,000 simulations.

bubbles kept prevalence throughout the course of the half-term (Fig 2A, green and yellow curves). Under all strategies, a proportion of asymptomatic infected individuals remained unidentified throughout the simulation (Figs 1D and 2B). By combining twice weekly mass testing with serial contact testing, the majority of asymptomatic individuals were identified over the course of the half-term. A lower proportion of asymptomatic individuals were identified when twice weekly mass testing was combined with isolation, as isolating individuals were not required to take LFTs.

## Absences

By isolating year group bubbles, even uninfected pupils can spend a considerable number of days absent. Under an isolation of year groups strategy, the average pupil spent 6.6 days isolating over the duration of the half term, i.e. around 19% of a seven week long half-term. Yet higher levels of absences resulted from a strategy of isolation combined with twice weekly mass testing, with the average pupil spending 15 days ($\approx$ 43% of school days) absent.

Strategies using LFTs that did not involve the isolation of year group bubbles resulted in much lower absences than strategies involving isolation. For serial contact tracing and mass testing strategies, as individuals would only be absent if they had sought a PCR test, or had tested positive to an LFT or PCR test, the majority of pupils had no days of school absence (95% under serial contact testing, 94% under twice weekly mass testing, 95% when combined). The mean days absent were 0.32 days for a serial contact testing strategy, 0.37 days for twice weekly testing, and 0.35 days for those measures combined (Fig 1E). Temporally, throughout the half-term under an isolation of year group bubble strategy a considerable portion of pupils (0–40%) may plausibly be expected to be absent (Fig 2C, orange curves). When combined with twice weekly mass testing, isolating year group bubbles led to very high levels of absences in the first week of term (60–80%), with high levels persisting throughout the term (Fig 2C, yellow curves). For the considered strategies that did not involve isolating bubbles, the fraction of students absent at any one time was relatively low, remaining below 2% throughout the half-term (Fig 2C).

## Testing demand

Strategies involving rapid testing required a large number of LFTs over the course of a half term. A mean number of 12 LFTs per pupil were required for a serial contact tracing strategy (Fig 1F, blue violin plot), though as a responsive measure the number of tests used varied considerably between simulations (95% PI: 4.7–18.8 LFTs per pupil). A higher number of LFTs were taken under a twice weekly mass testing strategy (15.8 LFTs per pupil, 95% PI: 15.7–15.9 LFTs per pupils) (Fig 1F, purple violin plot), and a yet higher number were taken for a strategy combining twice weekly mass testing with serial contact testing (27.6 LFTs per pupil, 95% PI: 24.4–30.6 LFTs per pupil) (Fig 1F, green violin plot). Fewer LFTs were required for a strategy of twice weekly mass testing when coupled with isolating year groups (Fig 1F, yellow violin plot), as pupils in isolation were not required to take LFTs (10.8 LFTs per pupil, 95% PI: (9.4–12.2 LFTs per pupil).

For strategies involving serial contact tracing, a high volume of testing was required throughout the half-term (Fig 2D). This high volume of testing also required many pupils to isolate over weekends, when tests were not administered.

## LFT participation

Our prior analysis considered situations where all pupils actively participate in the lateral flow testing required by the school's reopening strategy. Yet, pupils opting out of lateral flow testing

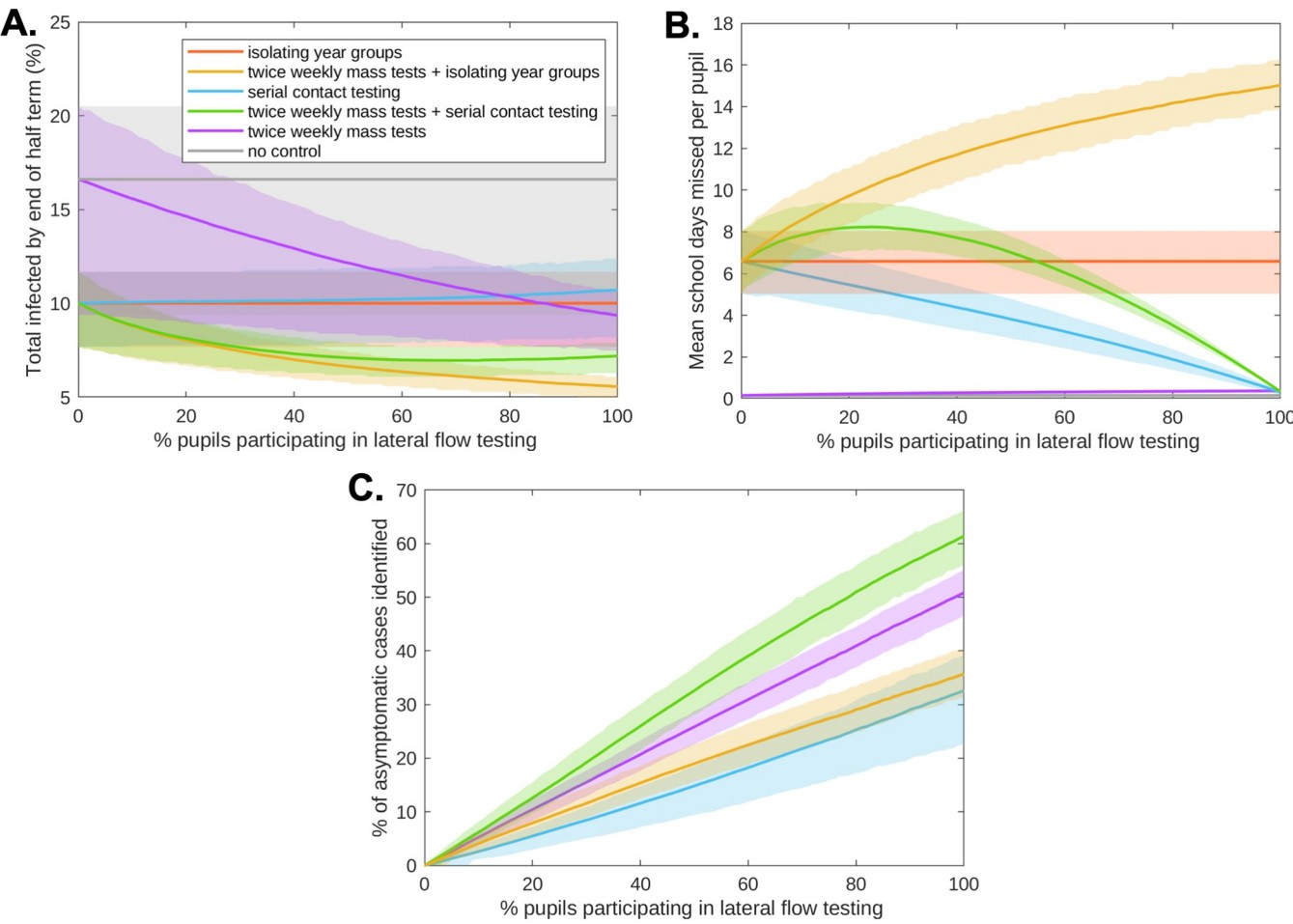

**Fig 3. The impact of pupil participation in lateral flow testing on infections and absences.** We varied the percentage of pupils who agreed to participate in lateral flow testing (0% to 100%, with 1% increments). **(a)** Total number of pupils infected by the end of the half-term. **(b)** Mean school days missed per pupil within a school over the course of the half-term. **(c)** For rapid testing strategies, the percentage of asymptomatic cases that had been identified through rapid testing by the end of the half term. In each panel, solid line traces correspond to the mean value attained from 2,000 simulations and shaded envelopes represent the 50% prediction intervals (these regions contain 50% of all simulations at each timepoint). The strategies displayed are: no control (grey), twice weekly mass testing (purple), serial contact testing (blue), isolating year group bubbles (orange), combined twice weekly mass testing and serial contact testing (green), combined twice weekly mass testing and isolation of year group bubbles (yellow).

can have a large impact on both transmission and absences (Fig 3). Twice weekly mass testing without further control measures required very high participation levels (> 86%) to be more effective (on average) than a strategy of isolating year group bubbles, and similarly required high levels (> 79%) to be more effective (on average) than a strategy of serial contact testing (Fig 3A). For strategies involving serial contact testing, since pupils opting out of lateral flow testing would be required to isolate upon identification of a positive case, at 0% pupil participation in lateral flow testing these strategies were equivalent to an isolation of year group bubbles strategy. Consequently, pupils opting out of lateral flow testing made a strategy *more effective* at reducing transmission, at the expense of resulting in a higher number of absences (Fig 3A). A strategy combining twice weekly mass testing and serial contact testing led to similar numbers of infections by the end of the half term when 30–100% of pupils participated in lateral flow testing. However, as a consequence of high levels of detection amongst participating pupils combined with the requirement of opted out pupils to isolate upon identification of a

positive case, if fewer than 54% of pupils participated in lateral flow testing a strategy combining twice weekly mass testing and serial contact testing led to a higher mean level of absences than the isolation of year group bubbles strategy (Fig 3B). The percentage of asymptomatic individuals identified increased with pupil participation for all strategies involving lateral flow testing (Fig 3C).

## Sensitivity analysis

As in any model, there is uncertainty surrounding the parametric assumptions that underpin its construction. Accordingly, we performed a univariate sensitivity analysis to understand the impact of these parameter assumptions on our findings (S2 Text). Across the range of alternative parameterisations considered, serial contact testing remained less effective at reducing infections than an isolation of year group bubbles strategy. This result is mirrored when both strategies are combined with twice weekly mass testing. While twice weekly mass testing was more effective at reducing infections than an isolation of year group bubble strategy under our baseline assumptions, this strategy became less effective than isolation when assuming a lower LFT sensitivity, and was also sensitive to the level of within-school transmission, probability of external infection, and assumed proportion of pupils symptomatic. However, twice weekly mass testing combined with serial contact testing remained more effective at reducing infections than an isolation of year group bubbles strategy. While levels of within-school transmission, probability of externally infection, and LFT sensitivity had the largest impact on the difference in absences between strategies, absences levels from strategies not involving the isolation of year groups remained markedly lower across the sensitivity assumptions considered.

As well as our univariate sensitivity analyses, we explored the specific impacts of: transmission on test days; levels of immunity among pupils; levels of within-school transmission; the impact of cohorting; and the impact of LFT user error. While allowing transmission on test days marginally increased the amount of cases over the course of the half-term, it did not impact the relative effectiveness of different control measures at reducing infections. Similarly, the relative effectiveness of control strategies was not impacted by the immunity levels amongst pupils at the beginning of the half-term, though at high levels of immunity only minor differences in the total number of infections by the end of the half-term were observed (irrespective of the control strategy that was applied). At high levels of within-school transmission, a strategy of twice weekly mass testing became less effective at reducing infections than either a strategy of isolating year group bubbles or serial contact testing. While the impact of cohorting itself was sensitive to whether one assumes frequency or density dependent transmission, the relative effectiveness of control strategies was not impacted by either assumption. If a large proportion of LFTs are taken incorrectly, reducing the sensitivity of such tests, a strategy of twice weekly mass testing combined with serial contact testing may become less effective at reducing infections than a strategy of isolating year group bubbles.

## Discussion

In this paper, we have developed an individual-based model of a secondary school formed of exclusive 'year group bubbles' and performed numerical simulations to assess the impact of a collection of postulated testing and isolation-based school control strategies (against spread of SARS-CoV-2) on transmission, absences and testing burden. Across the considered strategies, our findings reveal a trade-off between these three measures.

Evaluating control measures on the basis of school absences, strategies involving the isolation of year group bubbles led to substantially higher absences than strategies involving LFTs without isolation. We find that twice weekly mass testing can result in lower levels of infections

than a strategy of isolating year group bubbles, particularly when implemented in tandem with serial contact testing. We acknowledge, however, that such strategies require a high testing capacity. Prior work performing numerical simulations on complex networks has indicated a high volume of testing being required to effectively curb the spread of SARS-CoV-2 [64]. Combining twice weekly mass testing with the isolation of the year group was the most effective strategy (from the set we considered) at reducing infections, but resulted in very high levels of absences.

While serial contact testing considerably reduced absences compared to a strategy of isolating year group bubbles, serial contact testing remained slightly less effective than a strategy of isolating year group bubbles at controlling the number of infections. These results were echoed when both strategies were combined with twice weekly mass testing, and held true over the range of sensitivity assumptions considered. This result is echoed by the findings of a parallel study, considering the impact of LFT control strategies in the context of primary schools [44]. In comparison to isolating year group bubbles, thus breaking chains of transmission when a positive case is identified, rapid testing strategies can allow infected pupils who falsely test negative to continue to transmit infection within the school setting. However, the difference in infection levels over the half-term were relatively small, with the number of infections under serial contact testing 7% higher than the number under the isolation of year group bubbles. Accordingly, our results are broadly consistent with the results from a cluster randomised trial of secondary schools in England, which suggested serial contact testing to be non-inferior to the isolation of close contacts in reducing within-school transmission [31].

Under our baseline assumptions, and assuming all pupils participated, we found a strategy of twice weekly mass testing to be more effective than the isolation of year group bubbles at reducing infections. However, this required a high level of lateral flow testing participation by pupils to remain true. Further, the relative effectiveness of twice weekly mass testing compared to the isolation of year group bubbles was sensitive to a number of assumptions, including the level of within-school transmission, the probability of infection from the community, and the sensitivity of LFTs. Nevertheless, over the range of sensitivity assumptions considered a strategy combining twice weekly mass testing together with serial contact testing remained a more effective strategy at reducing infections than the isolation of year group bubbles.

Our results highlight that pupil participation in lateral flow testing is an important factor in its success, and demonstrate the potential unintended consequences of lateral flow testing strategies when participation levels are low. The finding of LFT based strategies leading to substantially fewer absences over the course of a half-term, compared to isolating year group bubbles of identified cases, was contingent on all pupils agreeing to participate in lateral flow testing. However, subsequent empirical studies indicate this may be an overly strong assumption; in a cluster randomised trial of serial contact testing of secondary schools in England, only 42.4% of identified contacts actively participated, when those not participating were required to self-isolate [31]. That being said, if levels of participation in lateral flow testing are low, we find that this requirement can negate the reduction in absences serial contact testing affords, with absences concentrated among pupils opted out of lateral flow testing. Through the detection of participating pupils, serial contact testing when combined with twice weekly mass testing can result in a *larger* number of absences than an isolation of year group bubbles strategy.

Previous modelling approaches have shown that the frequency of test screening has a larger impact on reducing transmission than test sensitivity [32]. However, while the sensitivity of LFTs had a smaller impact than levels of within-school transmission or community prevalence on infections over the half-term, its impact was sufficient to reorder the effectiveness of some of the strategies considered at reducing transmission. Differing model approaches between our study and the previously mentioned study [32] may affect the relative impact of test

sensitivity. While the study by Larremore *et al.* [32] assumed that test results were a deterministic function of viral load, in our model the likelihood of an individual testing positive was governed by a probability distribution dependent on the time since infection (inferred from observed data from UK healthcare workers [49]). These alternative approaches result in different levels of infectiousness removed through rapid testing, as under the assumptions of Larremore *et al.* individuals with high viral loads will always test positive to LFTs. This highlights the importance of continued research into the sensitivity of LFTs, with granularity to determine heterogeneity (if any) across specific age groups and in specific settings, as well as the most appropriate way to capture test probability profiles in a model. Our results demonstrate that the sensitivity of LFTs may still be an important determinant of the relative effectiveness of different school reopening strategies at reducing transmission.

While our model was parameterised to qualitatively reflect the epidemiological landscape in England in January 2021, when serial contact testing was suggested as a potential control strategy for secondary schools, it is important to consider the impact that the contemporary situation may have on the suitability of control strategies. Accordingly, our study has several limitations. The following paragraphs discuss the study's limitations regarding: (i) implications of vaccination of secondary school aged children, (ii) emergence of variants of concern, (iii) practicalities of testing, (iv) robust parameterisation of within-school contact structures, (v) applicability to primary schools, and (vi) risk to teachers and the wider community.

In England, because of the approval of a SARS-CoV-2 vaccine for all 12–17 year olds on the 13th September 2021 [65], combined with higher levels of natural immunity, substantially higher levels of immunity may persist among secondary school pupils at later periods in the pandemic. Our sensitivity analysis varying the initial level of immunity among pupils found that high levels of immunity did not impact the relative effectiveness of different control strategies at reducing infections. Yet, if high levels of immunity are achieved, the reduction in transmission from further measures may be slight and may still require high numbers of tests or result in high levels of pupil absences.

Second, the emergence of more transmissible variants may also impact the relative success of control measures. Assuming the infectivity profile of the virus is unchanged, we find that at high levels of within-school transmission, a strategy of twice weekly mass testing can become less effective than the isolation of year group bubbles at controlling transmission, but strategies combining twice weekly mass testing with either serial contact testing or isolation of year group bubbles remain effective over a wide range of levels of within-school transmission. We also stress that the SARS-CoV-2 infectivity profile [56] and the distribution of incubation periods [66] used in our analyses were derived from data collected from the wild-type strain of the SARS-CoV-2 virus. If these temporal profiles are different for a new variant, such that a larger proportion of cases occur when an individual is asymptomatic or presymptomatic, we anticipate that would impact the relative effectiveness of the studied strategies.

Third, our model makes several optimistic assumptions regarding the practicalities of testing. We assume that the sensitivity of LFTs that are self applied by pupils is comparable to that of LFTs self applied by healthcare workers [49], pupils required to isolate do so (and have no risk of becoming infected while isolating), and all symptomatic pupils seek a PCR test and isolate upon symptom onset. By doing so, we demonstrate the impact of reopening strategies if ideally implemented. We recognise this ideal scenario is unlikely to be met. If pupils do not take swabs correctly, and fewer cases are detected, this will increase the infections that occur when implementing a serial contact testing or regular mass testing strategy. Conversely, if pupils have a substantial risk of infection when they are supposed to be isolating, a strategy that keeps pupils within school, where social interactions are regulated, may become more beneficial. For strategies involving serial contact testing, pupils are expected to isolate on days

that they are due to be tested that fall on weekends. Pupils failing to isolate on serial contact test days that fall on weekends will reduce the effectiveness of these approaches.

Fourth, as we consider a secondary school implementing a bubbling strategy at the level of year groups, we make the simplifying assumption of random mixing within year groups, ignoring the heterogeneity in contact structure within year groups. The omission of contact heterogeneity at the individual level is common in individual-based models of SARS-CoV-2 transmission in schools [40, 41, 43, 44], and was necessitated by the time-constraints under which this work had to be conducted within to be useful during a pandemic. While studies prior to the COVID-19 pandemic have detailed contact patterns within schools [67–69], the implementation of stringent distancing measures within schools will have impacted such patterns. Some studies have used contact patterns measured before the COVID-19 pandemic [70], while other studies have used structured expert judgement to inform contact patterns [45] during the COVID-19 pandemic. However, data to robustly parameterise such patterns remains scarce, and it may be expected that within-school distancing measures have impacted the fragmentation and clustering of school contact networks. Going forward, contemporary surveys detailing contact patterns within schools and how these are affected by school-level distancing measures is an important line of research. Alongside these, modelling studies that assess the impact of heterogeneity and clustering in contact patterns on the effectiveness of school control strategies would be valuable. The modelling approach outlined in this paper could be extended to investigate both aspects.

Fifth, whilst this study has focused on the impact of reopening strategies in secondary schools, results may be expected to be qualitatively similar in the context of primary schools. Modelling studies focusing on primary schools [44, 45] have demonstrated the potential benefit of regular testing using LFTs in a primary school setting; these studies do, however, disagree about the potential of regular testing to control transmission within primary schools without the isolation of close contacts. Our modelling approach could be adjusted to the primary school setting by altering the size and number of year groups to instead reflect the size and number of classes in a primary school. However, there are some key differences that may impact the appropriateness of applying our results directly to a primary school setting. As tests within a school are expected to be self-administered, this may not be feasible for primary school age children, particularly in younger years. Epidemiological and clinical factors may differ between primary and secondary school aged children [71], meaning the effectiveness of serial contact testing, which often relies upon testing being initiated by a symptomatic case seeking treatment and testing positive, may be affected. The relevant size of the exclusive bubbles in primary schools will typically be smaller, as pupils are often partitioned into exclusive bubbles based on individual classes rather than entire year groups. While the use of smaller exclusive bubbles could potentially impact both transmission and testing, we find that the relative effectiveness of the control strategies considered in this manuscript is not impacted by group size.

Lastly, an evaluation of the risk of reopening strategies to teachers and the wider community is beyond the remit of this study, though this risk will be a function of the level of infections that result from different reopening strategies. Capturing this aspect of transmission would require the explicit modelling of teachers and external contacts of both pupils and teachers. Modelling approaches have been used to assess the impact of schools on wider transmission [72–74], but the data underlying these approaches depend on the control strategies implemented within schools. Other studies have attempted to capture this by including household contacts into an individual-based model of within-school transmission [40, 41]. Further research into the impact of testing strategies on transmission risk to the community would be a valuable line of enquiry for future research, but as is often the case, the challenge would be the appropriate parameterisation of the model.

By considering the level of absences realised and the volume of tests required under different reopening strategies, our study takes the first step in quantifying the indirect costs of SARS-CoV-2 transmission and control within schools. Future research should focus on incorporating the 'cost' of absences, tests, and cases amongst pupils into a health-economic framework, such as one that considers the impact of school control measures on quality adjusted life years (QALYs) [75]. Health economic modelling has been used to complement epidemiological models for a range of diseases and contexts, helping to inform policy [76–78]. However, a key challenge in implementing such an approach is quantifying the 'cost' associated with school days missed, how this varies across contexts, and to what extent this is mitigated by online learning. Further, other costs associated with school control strategies, such as the impact of control measures on the mental health of pupils, may be even harder to quantify. Studies prior to the COVID-19 pandemic have considered the impact of school days missed on attainment [79–81], while studies undertaken since the pandemic have attempted to quantify the impact of schools closures on loss of learning [82] and mental health [83]. Research synthesising existing knowledge into the harms associated with absences and control measures in the context of COVID-19, together with further research into these harms combined with epidemiological models of within-school transmission, is paramount to the design of optimal school control policies.

In summary, we have explored the impact of different reopening strategies on both transmission and absences within a secondary school setting. We find that although serial contact testing, acting alone, was less effective at reducing infections than an isolation of year group bubbles strategy, such a strategy can substantially reduce pupil absence from school. Acting alongside twice weekly mass testing, serial contact testing can reduce infections to levels lower than would occur under an isolation of year group bubbles strategy, but such a policy requires a high volume of testing. While a strategy combining twice weekly mass testing with isolation of year group bubbles was the most effective strategy considered at reducing infections, this strategy led to very high levels of absences. Our results highlight the conflict between the goals of minimising within-school transmission, minimising absences, and minimising testing burden. Amongst the strategies considered here we did not identify one that minimised all three aspects simultaneously. An assessment of the relative benefits and costs of each factor must be made when considering public health policies in response to SARS-CoV-2 transmission amongst school pupils.

## Supporting information

**S1 Text. Supporting methods.** Description of model algorithm and assumptions in detail. Descriptions of infectiousness over time, the parameterisation of within-school transmission, external infection, and recovery and immunity, extending the model to include interaction between year groups, and test probability profiles for symptomatic and asymptomatic individuals.
(PDF)

**S2 Text. Sensitivity analysis.** Description of the univariate sensitivity analysis performed, and the results from this analysis.
(PDF)

**S3 Text. Transmission on test days.** The impact of accounting for a probability of transmission between pupils on days in which they test positive, is explored.
(PDF)

**S4 Text. Increasing immunity levels among pupils.** The impact of higher levels of immunity among pupils on the effectiveness different reopening strategies is explored.
(PDF)

**S5 Text. Increasing within-school transmission.** The impact of higher levels of within-school transmission on the effectiveness different reopening strategies is explored.
(PDF)

**S6 Text. The impact of cohorting.** The impact of dividing year group into discrete non-interacting cohorts is on different reopening strategies is considered, exploring the assumptions of frequency and density dependent transmission in turn.
(PDF)

**S7 Text. The impact of LFT user error.** The impact of assuming that only a percentage of LFTs are taken correctly, thereby reducing overall test sensitivity, on the number of infections over the course of a half-term for strategies involving LFTs.
(PDF)

## Author Contributions

**Conceptualization:** Trystan Leng, Edward M. Hill, Robin N. Thompson, Michael J. Tildesley, Matt J. Keeling, Louise Dyson.

**Data curation:** Trystan Leng, Edward M. Hill.

**Formal analysis:** Trystan Leng.

**Funding acquisition:** Michael J. Tildesley, Matt J. Keeling, Louise Dyson.

**Methodology:** Trystan Leng, Edward M. Hill, Robin N. Thompson, Michael J. Tildesley, Matt J. Keeling, Louise Dyson.

**Software:** Trystan Leng, Edward M. Hill.

**Supervision:** Michael J. Tildesley, Matt J. Keeling, Louise Dyson.

**Validation:** Trystan Leng, Edward M. Hill.

**Visualization:** Trystan Leng.

**Writing – original draft:** Trystan Leng.

**Writing – review & editing:** Trystan Leng, Robin N. Thompson, Michael J. Tildesley, Matt J. Keeling, Louise Dyson.

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
