## [Decision Letter · Decision Letter 0]

28 Feb 2022

Dear Mr Leng,

Thank you very much for submitting your manuscript "Assessing the impact of lateral flow testing on within-school SARS-CoV-2 transmission and absences: a modelling study" for consideration at PLOS Computational Biology. As with all papers reviewed by the journal, your manuscript was reviewed by members of the editorial board and by several independent reviewers. The reviewers appreciated the attention to an important topic. Based on the reviews, we are likely to accept this manuscript for publication, providing that you modify the manuscript according to the review recommendations.

Sincerely,

Joseph T. Wu

Associate Editor

PLOS Computational Biology

Tom Britton

Deputy Editor

PLOS Computational Biology

[LINK]

Reviewer's Responses to Questions

**Comments to the Authors:**

Reviewer #1: This paper reports the results of a simulation study using an individual based model for the transmission of SARS-CoV2 within a school environment to compare and contrast the impact of alternative uses of rapid lateral flow tests in terms of the cumulative cases, absences and tests carried out.

The transmission model is necessarily largely assumptions based in the absence of useful empirical data at the school level to inform model development. Latent and infectious period distributions are based on the common set of empirical estimates from early in the pandemic used by most modelling studies.

The default configuration of the model assumes that transmission is well-mixed within year groups with limited exploration of alternative mixing patterns with assortative mixing between years and density versus frequency dependence (in supplementary information) and a range of different assumptions about prior immunity. Given the well mixed assumption the individual based structure is a convenience (compared to compartmental population models) rather than a necessity but will also likely have some computational performance advantages. Likewise, given this assumption the size of the school years (and school population) is the only specific parameter distinguishes the model from (smaller) primary schools.

The lack of an exploration of alternative transmission mechanisms is the largest limitation of the study. This is clearly described in the discussion, but I disagree with the justification (and necessity) for this assumption. We know from studies before the pandemic that social contact networks within schools are highly clustered - a network structure which is likely to limit and shape the rate of transmission within schools. The manuscript acknowledges this body of work but discounts the use of such data due to the likely changes in contact patterns due the pandemic. This is a valid point, however if anything social distancing measures might be expected to further fragment contacts and increase clustering. Network transmission scenarios (informed by empirical networks or simulated networks with similar patterns of clustering) would have provided a more robust evaluation of the sensitivity of results to model uncertainty and could very naturally be included within the chosen individual based modelling framework. Again, the time-critical nature of the policy relevance of such work makes such simplifying assumptions pragmatic (particularly given the lack of contemporary data) but it could be made clearer that such an exploration does not necessarily depend on carrying out time and resource consuming contact studies during a pandemic. The time-constraints under which work must be carried out to be relevant during a pandemic is a relevant and valid justification for the assumptions taken in itself and it would, in my view at least, be valuable to acknowledge it as such.

While parameter uncertainty is not, and cannot be, systematically explored given the type of data available, sensitivity of the results to parameter assumptions is addressed through a one at a time (OAT) analysis. This highlights the importance of the transmission parameters and test sensitivity which implies these parameters are likely to trade-off heavily against each other even if empirical data on transmission within schools was available.

The results are framed in terms of the trade-off between the number of infections averted, school absences and tests carried out. While the discussion acknowledges each of these have both a societal and economic costs the relative importance of each is not addressed which is vital for policy evaluation. Economic evaluations of public health control measures are routinely carried out using (albeit imperfect) measures such as DALYs. While I realise that there may not be reliable (or even any) quantitative estimates of these relative costs is this as they have not or cannot reasonably be evaluated. While the authors may well argue that this is a question of economics rather than biology, the policy implications of this work hinges on the answer whether this is a personal value judgement or something that can be more systematically quantified.

To sum up, this is a well written account of a careful assumptions based modelling exercise to explore the trade-offs between different testing strategies in schools to limit the impact of SARS-CoV and associated control measures. The methods are clearly documented both within the paper and with additional detail in supplementary information. Full source code to replicate the analysis is linked and a reasonable attempt at exploring parametric and model sensitivity of the results has been carried out. Necessarily carried out under time constraints and used as part of the package of advice given to the UK government this work has already demonstrated impact and it's publication is valuable in itself with respect to adding to the public record of the response to the ongoing pandemic.

Minor comments

Introduction: While it is clearly a NPI, practicing good hand hygiene is not, unlike the other listed NPIs, really a social restriction and not as far as I know enforced legally in the same way. In the context of the point being made it's inclusion at the start of the list distracts a little from the main point being made.

Reviewer #2: Thank you for this interesting simulation study on different variations of repetitive testing in schools.

The results would be more generalizable to other settings if a) it was not assumed that student participation was 100% in the main analysis (seems unlikely in most settings), and b) the error rates of the tests were adjusted to account for user error (both due to imperfect testing procedures, and possible changes due to changing variants).

In Figure 3, it would have been interested to see also the number of missed infections.

Thank you for pointing out in the limitations that these results are actually quite optimistic. It would seem that "user error" in the testing with LFT is probably somewhat high in this population. Were any simulations performed assuming higher error rates?

**Have the authors made all data and (if applicable) computational code underlying the findings in their manuscript fully available?**

Reviewer #1: None

Reviewer #2: Yes

PLOS authors have the option to publish the peer review history of their article (what does this mean?). If published, this will include your full peer review and any attached files.

Reviewer #1: No

Reviewer #2: **Yes: **Sarah R Haile

Figure Files:

Data Requirements:

Reproducibility:

References:

---

## [Decision Letter · Decision Letter 1]

2 May 2022

Dear Mr Leng,

We are pleased to inform you that your manuscript 'Assessing the impact of lateral flow testing strategies on within-school SARS-CoV-2 transmission and absences: a modelling study' has been provisionally accepted for publication in PLOS Computational Biology.

Best regards,

Joseph T. Wu

Associate Editor

PLOS Computational Biology

Tom Britton

Deputy Editor

PLOS Computational Biology

Reviewer's Responses to Questions

**Comments to the Authors:**

Reviewer #1: Many thanks for comprehensively addressing my comments and suggestions.

Reviewer #2: No further comments.

**Have the authors made all data and (if applicable) computational code underlying the findings in their manuscript fully available?**

Reviewer #1: Yes

Reviewer #2: Yes

PLOS authors have the option to publish the peer review history of their article (what does this mean?). If published, this will include your full peer review and any attached files.

Reviewer #1: No

Reviewer #2: No

---

## [Editor Report · Acceptance letter]

24 May 2022

PCOMPBIOL-D-21-02289R1 

Assessing the impact of lateral flow testing strategies on within-school SARS-CoV-2 transmission and absences: a modelling study

Dear Dr Leng,

I am pleased to inform you that your manuscript has been formally accepted for publication in PLOS Computational Biology. Your manuscript is now with our production department and you will be notified of the publication date in due course.

With kind regards,

Anita Estes
